# Smart Bio-Nanocoatings with Simple Post-Synthesis Reversible Adjustment

**DOI:** 10.3390/biomimetics10030163

**Published:** 2025-03-07

**Authors:** Mikhail Kryuchkov, Zhehui Wang, Jana Valnohova, Vladimir Savitsky, Mirza Karamehmedović, Marc Jobin, Vladimir L. Katanaev

**Affiliations:** 1Department of Cell Physiology and Metabolism, Faculty of Medicine, University of Geneva, Rue Michel Servet 1, CH-1211 Geneva, Switzerland; 2School of Chemistry and Pharmaceutical Engineering, Shandong First Medical University (Shandong Academy of Medical Sciences), Tai’an 271016, China; 3Zoological Museum, Moscow Lomonosov State University, Bol’shaya Nikitskaya Street 2, Moscow 125009, Russia; 4Department of Applied Mathematics and Computer Science, Technical University of Denmark, DK-2800 Kgs. Lyngby, Denmark; 5Haute ecole du Paysage, D’ingenierie et D’architecture de Geneve, University of Applied Sciences of Western Switzerland (HES-SO), 4 Rue de la Prairie, CH-1202 Geneva, Switzerland

**Keywords:** protein-based coatings, self-assembly, active coating, smart materials, bio-mimetic, switchable, antireflective, eco-friendly

## Abstract

Nanopatterning of signal-transmitting proteins is essential for cell physiology and drug delivery but faces challenges such as high cost, limited pattern variability, and non-biofriendly materials. Arthropods, particularly beetles (Coleoptera), offer a natural model for biomimetic nanopatterning due to their diverse corneal nanostructures. Using atomic force microscopy (AFM), we analyzed Coleoptera corneal nanocoatings and identified dimpled nanostructures that can transform into maze-like/nipple-like protrusions. Further analysis suggested that these modifications result from a temporary, self-assembled process influenced by surface adhesion. We identified cuticular protein 7 (CP7) as a key component of dimpled nanocoatings. Biophysical analysis revealed CP7’s unique self-assembly properties, allowing us to replicate its nanopatterning ability in vitro. Our findings demonstrate CP7’s potential for bioinspired nanocoatings and provide insights into the evolutionary mechanisms of nanostructure formation. This research paves the way for cost-effective, biomimetic nanopatterning strategies with applications in nanotechnology and biomedicine.

## 1. Introduction

Nanopatterning of signal-transmitting proteins plays an essential role in cell and tissue physiology and metabolism, from mechanotransduction [1,2,3] to context-dependent and reconfigurable signaling [4]. These properties of the nanopatterned surfaces are becoming even more critical with the increasing usage of nanostructured cargo microparticles [5,6]. Such microparticles are very promising in drug delivery, and enhancing their activity and specificity is essential for improving diagnostic and therapeutic outcomes [7,8,9,10]. Recent advancements in protein immobilization strategies, such as photocatalytically polymerized brushes [11], have led to spatially defined protein adhesion. Techniques that use noninvasive methods, such as light-sensitive photocatalysts and hydrogels [12,13,14] on surfaces with well-defined microenvironments, addressing challenges in cell biology and tissue engineering are crucial for developing biosensors and protein microarrays. Despite its potential, nanopattern technology faces many challenges, such as manufacturing and designing nanoscale geometries [15], reproducibility, and high production costs [16] with relatively low yields [1]. Current technologies of surface nanopatterning and nanostructured particle synthesis are limited in the range of patterns they can produce and often rely on non-biocompatible materials and techniques [17]. Therefore, biomimicry offers a promising approach to developing more efficient, cost-effective, and bio-friendly nanopatterning strategies [18,19,20].

One key factor for arthropods’ evolutionary success is their natural mastery of nanotechnology. The surfaces of most arthropods feature various nano-sized outgrowths, grooves, or depressions, which provide functional advantages such as improved vision or self-cleaning [21,22]. These enhanced properties arise from significant changes in the physical and chemical properties of materials at the nanoscale. It was shown that these nanostructures form by the self-assembly process of waxes and proteins. This discovery has enabled the reproduction of the self-assembly process in vitro [23], allowing for the modification of different surfaces. Since these nanocoatings are protein-based, they can be engineered to create surfaces involved in signaling or enzymatic activity. However, such modifications can disrupt the protein self-assembling ability [24]. To overcome this challenge, an alternative approach based on evolutionary analysis might provide a solution [18]. Based on these initial observations, we conducted a large-scale screening of the Coleoptera order to identify trends and patterns of potential solutions.

## 2. Materials and Methods

Coleoptera samples were obtained from the Zoological Museum of Lomonosov Moscow State University (Moscow, Russia). The cuticles from a minimum of 3 animals of each species were carefully excised. The samples were then mounted on coverslips using double-sided adhesive tape. Samples were stored in closed Petri dishes, and it was shown that they are stable in air for at least 4 years [23]. For the wettability tests, some samples were washed in Milli-Q water while rubbing with a cotton swab.

### 2.1. Atomic Force Microscopy (AFM)

Atomic force microscopy (AFM) was performed using multiple instruments and operational modes. Measurements in tapping mode were conducted with a Shimadzu SPM-9600, using NS16 cantelevers (tip radius 8 nm, frequency 190 kHz, length 225 μm) (NT-MDT, Moscow, Russia). Imaging in contact mode (0.3 nN) was performed using an XE-100 microscope (Park Systems, Suwon, Republic of Korea) using PPP-NCHR cantilevers (tip radius of curvature < 10 nm, length 125 μm) (NanoWorld, Neuchâtel, Switzerland). Adhesion Force Microscopy was performed by the NanoWizard 4XP (Brucker, Billerica, MA, USA) in QI mode using Scansyst-air cantilevers (tip radius 2 nm, frequency 70 kHz, length 115 μm) (Brucker). The Gwyddion software version 2.67, developed by the Department of Nanometrology at the Czech Metrology Institute (Brno, Czechia), was employed to visualize and quantify the acquired data.

### 2.2. Simulations

Simulations were performed using the following approach: the surface roughness was characterized in terms of the root means square (RMS) roughness, *σ* defined by σ=a−2∫x=−a2a2∫y=−a2a2hx,y2dydx, where *a* is the side length of the square sample, and *h*(*x*,*y*) is the function describing the height of the sample surface at (*x*,*y*). For the considered range of wavelengths *λ* (300 nm to 800 nm), we find that the ratio *σ/λ* is bounded by approximately 0.1 for all AFM scans, which is a sufficiently small value to justify the use of the total integrated scatter (TIS) formula for reflectance at normal incidence [25], $R\left(\lambda \right)={\left|\frac{n_{\mathrm{substrate}}-n_{0}}{n_{\mathrm{substrate}}+n_{0}}\right|}^{2}exp\left(-16\pi^{2}\sigma^{2}/\lambda^{2}\right)$. Here, *n*_substrate_ = 1.6 [26] and *n*_0_ = 1 are the refractive indices of the substrate material and of free space, respectively, and the squared fraction in the reflectance formula is the diffraction efficiency for a smooth air–substrate interface at normal incidence.

### 2.3. Wettability Test

To assess surface wettability, Milli-Q water droplets were dispensed using a capillary attached to a micromanipulator (MN-4, MMO-203, Narishige, Tokyo, Japan). The behavior of the water droplets was observed under an inverted microscope (Axiovert 40 C, Zeiss, Oberkochen, Germany). Images of the droplets were captured using a digital camera, and the contact angles of the droplets were measured on both sides using Gwyddion software version 2.67.

### 2.4. Mass-Spectrometry

Corneal samples for proteomic analysis were prepared by excising the eyes from the heads of mature adult beetles using a scalpel. The excised eyes were scraped from the inner surface and washed three times with Milli-Q water. The cleaned corneal samples were then incubated at 95 °C in loading buffer (62.5 mM Tris-HCl, pH 6.8, 10% glycerol, 2% SDS, 1% β-mercaptoethanol, a trace amount of bromophenol blue) for 2 h, as previous experiments [23,27] have shown that this time is acceptable to wash a sufficient amount of the material from the lenses but not long enough to lead to protein degradation. The samples were analyzed using SDS-PAGE, and the bands corresponding to the protein of interest were excised and sent to the Protein Analysis Facility of the University of Lausanne (Lausanne, Switzerland). Proteins were in-gel digested with trypsin, and peptides were analyzed by nanoLC-MSMS using an easy-nLC 1000 liquid chromatography system (Thermo Fisher Scientific, Waltham, MA, USA) coupled with a Q-Exactive HF mass spectrometer (Thermo Fisher Scientific, Waltham, MA, USA).

### 2.5. Protein Purification

For protein purification, the gene sequence encoding the CP7 protein with an RGS-His tag at the N-terminus was custom-synthesized and cloned into a pET-28a(+) vector (Synbio Technologies, Monmouth Junction, NJ, USA). The plasmid was transformed into the *E. coli* BL21(DE3)pLysS strain for recombinant protein expression upon induction with 1 mM IPTG. The bacterial mass was lysed by a cell press (Constant Systems, Daventry, UK). The His-CP7 was purified using the HisPur Ni-NTA resin (ThermoFisher Scientific, Waltham, MA, USA) with prewashing using a 2 M Urea solution to exclude the possibility of protein–protein binding. This was followed by purification using the Äkta liquid chromatography system using the Superdex^®^ 200 Increase 10/300 GL column (Sigma-Aldrich, St. Louis and Burlington, MA, USA).

### 2.6. Sodium Dodecyl Sulfate–Polyacrylamide Gel Electrophoresis (SDS-PAGE)

For Sodium Dodecyl Sulfate–Polyacrylamide Gel Electrophoresis (SDS-PAGE), proteins were mixed with Laemmli buffer, boiled at 95 °C for 5 min, and loaded onto an 18% acrylamide gel. For Coomassie staining, the gels were washed twice with water for 5 min, then incubated in colloidal Coomassie stain solution for 2 h. Destaining was performed overnight in water.

### 2.7. Western Blotting

For Western blotting, proteins were transferred from the gel to a polyvinylidene difluoride (PVDF) membrane (GE Healthcare, Chicago, IL, USA) using a wet transfer system (Bio-Rad, Hercules, CA, USA) at 100 V for 3 h. The membrane was then blocked in 5% low-fat milk in TBS for 1 h, which is a standard procedure used to reduce the background signal on the membrane, followed by incubation with an anti-RGS-His primary antibody (mouse, 1:1000 dilution, Qiagen, Hilden, Germany, cat. no: 34650). HRP-conjugated secondary antibodies were used for detection via enhanced chemiluminescence in a Fusion FX6 Edge system (Vilber, Collégien, France).

### 2.8. Wax Emulsion Preparation

Wax emulsions were prepared by adding 4 g of lanolin wax (Sigma-Aldrich, St. Louis and Burlington, MA, USA) to 40 mL of 10% SDS solution and sonicating the mixture at 80 °C for 2 h in a water bath (AL 04-04, ThermoFisher Scientific, Waltham, MA, USA) [28,29]. The emulsified solution was left to stand at room temperature for 24 h, after which the upper layer enriched in wax nanodroplets was collected, diluted tenfold in 1× TBS, and incubated for 48 h. After incubation, only the lower layer of the solution was collected and diluted in TBS until it reached a concentration equal to 0.8 mg/mL (for CD) or 0.4 mg/mL (for artificial coatings). The final verification was carried out using a ZetaSizer Nanoseries (Malvern Pananalytical, Malvern, UK) analyser, using DTS 1070 cells (Malvern, UK) according to the manufacturer’s protocol.

### 2.9. Circular Dichrosim

Circular dichroism measurements were performed in the Department of Organic Chemistry of the University of Geneva using the Jasco J-815 circular dichroism spectrometer with strain-free QS quartz 1 mm path-length cuvettes (Jasco, Easton, MD, USA). The His-CP7 protein was used at 0.23 mg/mL in 25 mM potassium phosphate buffer, pH 7.5.

### 2.10. ThT Fluorescence Measurement

The ThT fluorescence levels were measured by the Multimode plate reader Hidex Sense (Hidex, Turku, Finland) from the plate (microplate 96 well, Greiner Bioone, Kremsmünster, Austria) bottom with excitation of 355 ± 40 nm and emission of 535 ± 20 nm [30]. The final concentrations were 30 μM ThT (Sigma-Aldrich, St. Louis and Burlington, MA, USA, #T3516), 0.12 mg/mL proteins, and 0.004% SDS. Before measurement, ready solutions were incubated for 10 min at room temperature.

### 2.11. Nanocoating Generation

Nanocoatings were created by applying a mixture of 5 μL His-CP7 (0.65 mg/mL in TBS) and 15 μL of lanolin (0.4 mg/mL in TBS) emulsion onto a 1 cm^2^ glass coverslip. The applied mixture was dried, rinsed with water, and re-dried. This process was repeated twice to ensure proper coating. Switching of corneal or artificial nanocoatings was performed by applying a solution of 1 μL His-Reinin (0.5 mg/mL in TBS), 2 μL of lanolin (0.4 mg/mL in TBS) emulsion, and 16 μL of TBS on surfaces. The applied mixture was dried, rinsed with water, and re-dried.

### 2.12. Statistical Analysis

Statistical analysis was performed using GraphPad Prism software Version 10.0.3.

## 3. Results

### 3.1. Two Types of Nanostructures and Their Functionality

Nanoscale analysis of the corneal structures in Coleoptera revealed variations in nanocoatings among different specimens [24]. Atomic force microscopy (AFM) analysis of the superfamily Tenebrionoidea showed that the commonly observed dimpled nanocoatings are sometimes modified into maze-like or nipple-like structures (Figure 1a–d) [21,31]. This duality was observed in species such as *Melandrya caraboides*, *Oedemera virescens*, *Schizotus pectinicornis*, *Rhampholyssa steveni*, *Mylabris calida*, and the *Omophlus* sp. (Figure 1a–c). These structures resemble the surface textures found in butterfly eyes [32]. Such modifications are thought to enhance antireflectivity, which is considered the primary function of corneal nanocoatings [33]. The reduction in reflected light is attributed to the formation of a refractive index gradient by nanostructures smaller than the wavelength of the light. The effective refractive index at any depth of such nanocoating is the weighted sum of the refractive indices of the materials present. For instance, if half of the coating consists of lens material and the other half air, the effective refractive index will be between those of air and the lens material [34,35]. Light passes through this nanostructured coating as if through a medium with a smooth gradient of refractive index. Numerous theoretical and experimental investigations into nanostructures have shown that taller structures tend to exhibit superior antireflective properties [35,36], suggesting that the nipple-like and maze-like structures provide better antireflective properties. However, these coatings are also more prone to damage, which can compromise their functionality. This creates a dilemma: how to maximize efficiency while minimizing the risk of environmental harm. For example, flying insects like butterflies can support higher nanopillars [37], as they are less susceptible to collisions compared to crawling insects [38].

Due to the uneven distribution of these modified nanostructures, we could not experimentally verify this hypothesis. To address this limitation, we decided to simulate the light reflectance using the total integrated scattering method. The results showed a significant reduction in light reflectance for both tested insect specimens (Figure 1e).

Nanocoatings have a significant effect on the wettability of the materials they cover. There are two primary models that describe this behavior: the Wenzel and the Cassie–Baxter models. The Wenzel model assumes that the entire surface is completely wetted, while the Cassie–Baxter model posits that gas becomes trapped in the cavities created by the surface’s roughness. Experimental data reveal that the actual interactions between solid, liquid, and gas often fall somewhere between these two models, influenced by the morphology of the structures involved [38,39]. Usually, structures smaller than 50 nm exhibit wetting behavior as described by the Wenzel model, which largely depends on their chemical composition [26,40]. Since in the Wenzel state the antireflective and antiadhesive functionalities of corneal nanocoatings are typically mutually exclusive [23], we tested the hydrophobicity of these surfaces. As expected, the modification made the surface hydrophilic, as indicated by a drastic decrease in the contact angle to approximately 70°, a characteristic value for a standard nipple array structure [24]. However, the nature of this modification was temporary, suggesting that it could be washed away along with other contaminations if needed [41]. Indeed, gentle washing with water, combined with light rubbing, partially restored the hydrophobicity (Figure 1f).

### 3.2. Switching Mechanism

Given the rigid nature of the corneal surface, we proposed that the structural changes could result from a specific form of directed contamination. As previously shown, grooves of nanocoatings differ in nature compared to the hills [42,43]. With an ester-based chemical composition, the grooves exhibit higher adhesion, which is reflected in their better wettability [23,40,44]. To confirm this, we directly measured the adhesion of *R. steveni* corneal samples by using AFM. Indeed, the results revealed that the composition and, consequently, the adhesion of the nanocoating is not uniform. The maximal adhesion force was colocalized with topographical dimples (Figure 2a,b). We suggested that these dimples in the corneal coating could direct the next step of the assembly, facilitating the growth of new, secondary nano-protrusions on the top of the hollows and dimples (Figure 2c). These secondary nanostructures are less stable and can be easily washed away, but, at the same time, they might enhance the antireflective properties of the dimpled structures.

The switching mechanism might involve the secretion of a specific adhesive material. For example, the same self-assembling protein/wax mixture could be applied on a surface, as was shown for whip spiders [17]. To test this hypothesis, we artificially coated the dimpled corneal cuticle of *Tribolium castaneum* (Tenebrionoidea) with a diluted solution of wax and protein. This solution was able to form protrusions on top of existing nanodimples (Figure 2d,e).

### 3.3. Reverse Engineering

In our previous work, we developed a workflow that enabled the identification of Retinin as the self-assembling morphogen in the corneal nanocoatings of *Drosophila* [23]. Using the same approach, we analyzed the corneae of other insects with fully sequenced genomes, including the silk moth *Bombyx mori* [27], the common Mormon butterfly *Papilio polytes* [26], and the mosquito *Anopheles gambiae* [45]. Given that the red flour beetle (*T. castaneum*), a standard representative of the Tenebrionoidea superfamily, also has a fully sequenced genome [46], we investigated its corneal proteomics (Figure 3a). Despite challenges in the material collection [47], we successfully identified the primary small corneal protein as TC003109, also known as cuticular protein 7 (CP7).

CP7 from *T. castaneum*, Retinin from *D. melanogaster*, CPR10 from *A. gambiae*, CPR67A from *P. polytes*, CPR150 and CPR19 from *B. mori* are all small secreted proteins that lack sequence homology with one another and are predicted to have extensive intrinsically disordered regions (IDRs). The disordered nature of Retinin and CPR67A has been confirmed in our previous work [23,26]. A critical feature enabling these proteins to participate in nanocoating formation is the lipid-binding capacity of their IDRs. Indeed, many IDRs are known to interact with lipids [48], and lipid-binding activity may be a common characteristic of IDRs and IDR-containing proteins [49,50,51,52,53,54]. Not only corneal proteins but also other IDR-containing proteins have been shown to adopt an induced-fit conformation upon interacting with lipids [54]. The formation of high-molecular-weight condensates composed of IDR proteins and lipids has also been experimentally observed [49], suggesting that it could be the first step in the nanostructure self-assembly process [23]. Analysis using the DisoLipPred algorithm [50] confirmed the presence of lipid-binding regions in CP7, CPR67A, CPR10, revealing a similar organization of these regions across the proteins (Figure 3b). Based on these findings, we propose that CP7 is responsible for the formation of Tenebrionoidea-like dimpled corneal nanocoatings. However, recombinant CP7 protein was found to behave differently than previously investigated CPR10 and CPR67A. Size-exclusion liquid chromatography, Coomassie-stained SDS-PAGE, and Western blot analyses of recombinant His-CP7 showed that this protein exists as dimers and oligomers, even under denaturing conditions (Figure 3c). Previous data on protein purification using this method [24] and an additional step of washing Ni-NTA beads exclude the possibility of contamination. Indeed, in silico [55] analysis indicated that CP7 dimers could form in solution without any ligands. Adding myristic acid to this simulation resulted in additional folding and the assembly of a more compact complex (Figure 3d).

Biophysical analysis of CP7 confirmed its distinct properties, differentiating it from the group of previously identified nanocoating-forming proteins. Circular dichroism analysis has shown that CP7 is initially folded, and its interaction with wax alters its tertiary structure but not as drastically as in the case of other Retinin-like proteins (Figure 4a). At the same time, we observed the absence of β-sheets capable of binding ThT [56] in the protein without a wax, which appeared gradually as the wax concentration increased. However, the intensity of the interaction with ThT was even lower than that for the least active CPR67A protein (Figure 4b).

Given the unique properties of CP7, we proceeded to test the self-assembly of the CP7/lanolin wax mixture on the glass. Using the protocol that produced nipple array nanocoatings in the case of CPR10 and CPR67A, we successfully reproduced the dimpled pattern with the CP7 protein. Additionally, secondary coating of the dimpled surface with a diluted wax and protein solution resulted in the formation of protrusions on top of existing nanodimples, similar to the corneal samples (Figure 4c).

## 4. Discussion

The evolution, origin, and development of modifiable nanocoatings on the insect cuticle remain enigmatic. While all studied beetles belong to the same superfamily of Tenebrionoidea, their classification provides little insight into the specific evolutionary pressures of mechanisms behind the emergence of this unique adaptation. Furthermore, the ecological diversity of these beetles, both as adults (imago) and larvae, varies significantly among the four studied species and complicates the search for a unifying driver. Adults of *O. virescens* inhabit meadows, ranging from forested areas to the steppe zone, and are typically found on flowers [57]. In contrast, *R. steveni* occupies sandy and floodplain areas in semi-desert and desert regions, and adults are also commonly found on flowers [58]. *M. caraboides* and *S. pectinicornis* are forest-dwelling species, but their specific habitats differ [59]. Adults of *S. pectinicornis* are primarily found on tree trunks, particularly birch [60], while *M. caraboides* reside on trunks and beneath the bark of dead trees [61]. These ecological variations in these species do not suggest a specific evolutionary path driven by specific selective pressures. It is possible that these nanodimpled coatings are primordial and appeared earlier than the Tenebrionoidea superfamily, as similar structural modifications have been observed on the surface of underwater amphipods [62]. Their bi-functionality possibly provided just the right ecological trade-off to persist in the lineages of arthropods.

There are three potential mechanisms by which insects modify their corneal coatings. The first involves the use of naturally occurring impurities deposited on the surface from various sources. However, this mechanism has significant limitations, as the physical properties of these materials are inherently unpredictable. The second mechanism, discussed in the results, involves the secretion of a mixture capable of self-assembly on the dimpled surface. We suggest that such a mixture can consist of compounds similar to those that are assembled by the reaction–diffusion mechanism to the initial nanocoatings. And reversibility is then easy to achieve by limiting the binding of newly appearing nanostructures to the surface. While such behavior has been observed in whip spiders [63], there are currently no reports or observations of beetles distributing secretion material over the surface of their eyes. It is also possible that this secretion is produced by hypodermal cells within the eye cuticle. The secretion would then exit onto the eye’s surface through pore channels. Although such channels have been identified in some hexapods [64] and in olfactory organs [31], no evidence of pore channels has been reported for insect corneas [37,65,66].

The antireflective functionality of these modifications can exist only in the case of the dimpled pattern. Appearing protrusions double the distance from hollows to the peaks, which is the main factor in decreasing light reflectance [35,36]. In contrast, the nipple array would only be flattened by such modifications, as was shown for *D. mojavensis* and *P. delalandei* [23,26]. However, the presence of such flattened modified nanocoatings lets us suggest the existence of additional functionality, such as anti-contamination properties against waxy particles or against dust, as shown in the case of whitefly [67].

The newly identified protein CP7 differs from all other Retinin-like proteins, possessing a stronger tendency for the formation of oligomers and much weaker activation by wax simultaneously. The higher self-activation rate and lower activation by the inhibitor could explain this phenomenon. These data not only provide a way to detect the exact molecular mechanism of protein-based nanostructure assembly but also allow for the combination of different building blocks, as demonstrated in this article.

## 5. Conclusions

This study highlights that the dimpled pattern plays a crucial role in antireflective properties, reducing light reflectance, and may serve additional purposes, such as anti-contamination, due to its ability to guide the secondary reversible coating by its specific topographical and adhesive characteristics.

The artificially engineered structures we developed closely replicate the physical properties observed in natural systems, demonstrating the feasibility of their synthesis under mild and controlled conditions. This innovative approach offers significant advantages, particularly for applications where conventional treatment methods are unsuitable, such as the fabrication of contact lens surfaces, living tissues, or prosthetic devices. For these surfaces, the limitations of coatings, such as relatively weak binding to the substrate and low wear resistance [45], are minor.

These synthetic coatings consist of a wax-enriched base layer that provides durability, complemented by a protein-enriched layer on the top. The unique construction of these coatings opens up possibilities for a wide range of applications, extending their utility beyond the original purpose. For example, proteins in the modification layer can be engineered with peptides or short proteins and then nanopatterned through a self-assembly mechanism. This approach enables the precise programming of structure density and size, which has been shown to enhance protein stability [68]. Surfaces with pre-patterned active proteins could lead to the development of novel sensors as well as advancements in tissue engineering [69] and the study of cell behavior [70].

## Figures and Tables

**Figure 1 biomimetics-10-00163-f001:**
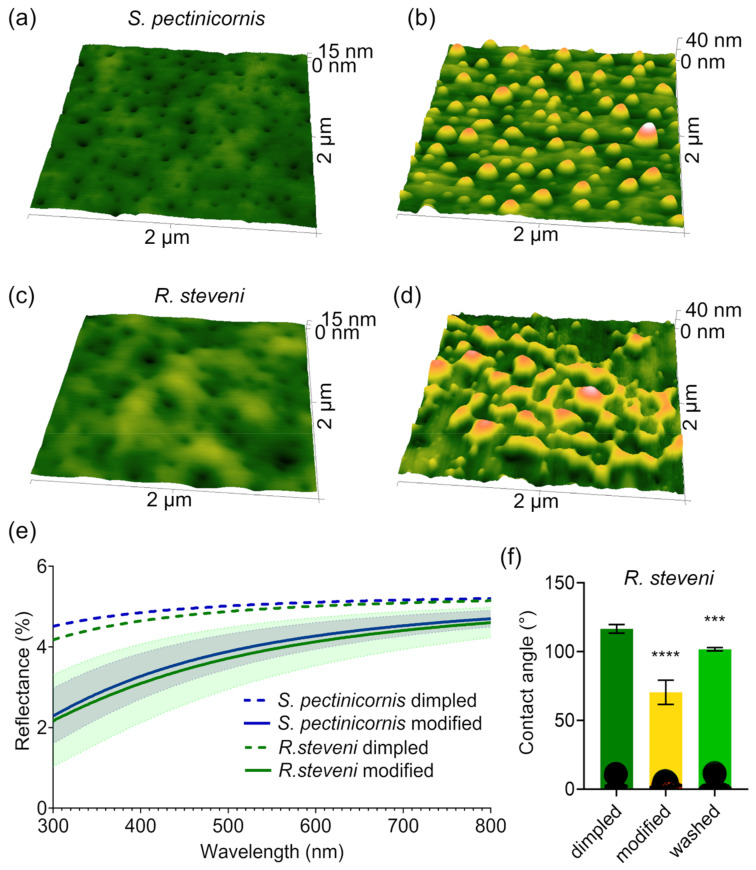
Unmodified and modified corneal nanocoatings. (**a**,**d**) Unmodified (**a**,**c**) and modified (**b**,**d**) corneal nanocoating AFM topography scans of *S. pectinicornis* (**a**,**b**) and *R. steveni* (**c**,**d**). (**e**) Simulated reflection spectra of the investigated surfaces. Clear improvement of the antireflectivity by modification is visible. Data are shown as mean (lines) ± range (semi-transparent). (**f**) Water contact angles of the eye surface from *R. steveni* unmodified (dimpled) samples, modified samples, and modified samples after cleaning. The bar and error bars indicate the mean ± SD; n = 4. Statistical significance is tested by a two-tailed *t*-test (“***” *p* < 0.001, “****” *p* < 0.0001). The representative images of the droplets are shown as bar inserts.

**Figure 2 biomimetics-10-00163-f002:**
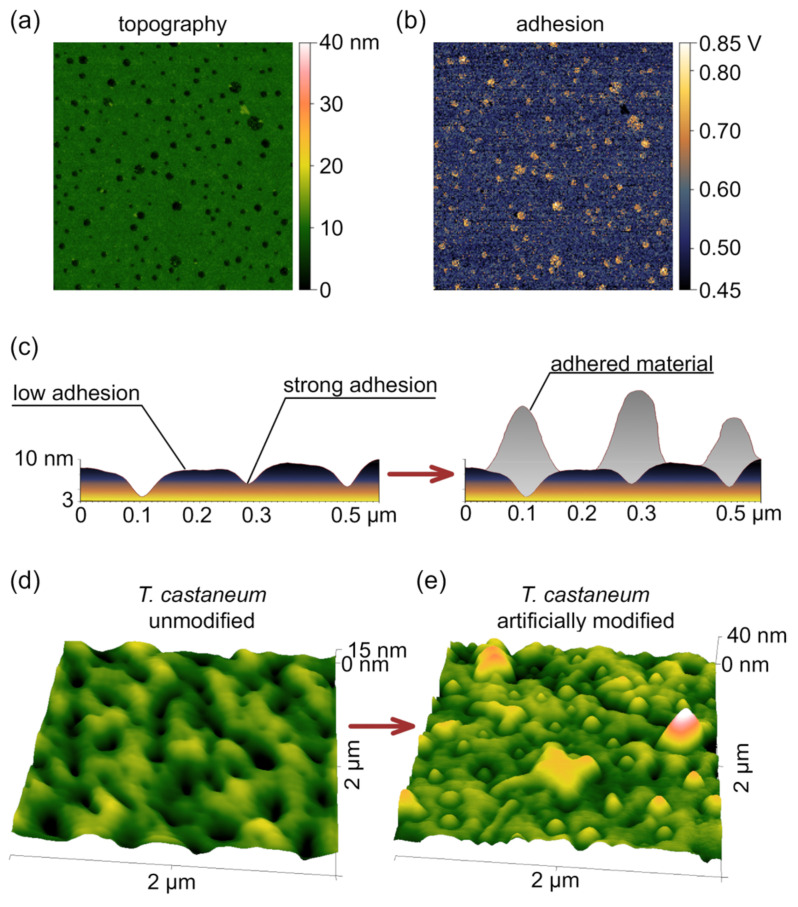
The uneven adhesion and topography of Tenebrionoidea corneal nanocoatings lead to area-specific deposit aggregation. (**a**) AFM topography scan of the *R. steveni* cornea. (**b**) The same area, but showing adhesion data on the nanoscale. Clear topography dependence is visible. Scans are 2 × 2 µm. (**c**) Proposed mechanism of a specific material aggregation over the most adhesive areas. Heights and shapes are based on AFM topographical data of *R. steveni* corneae. (**d**) *T. castaneum* corneal nanocoating AFM topography scan. (**e**) After applying a self-assembling wax–protein mixture, the same sample shows the appearance of protrusions.

**Figure 3 biomimetics-10-00163-f003:**
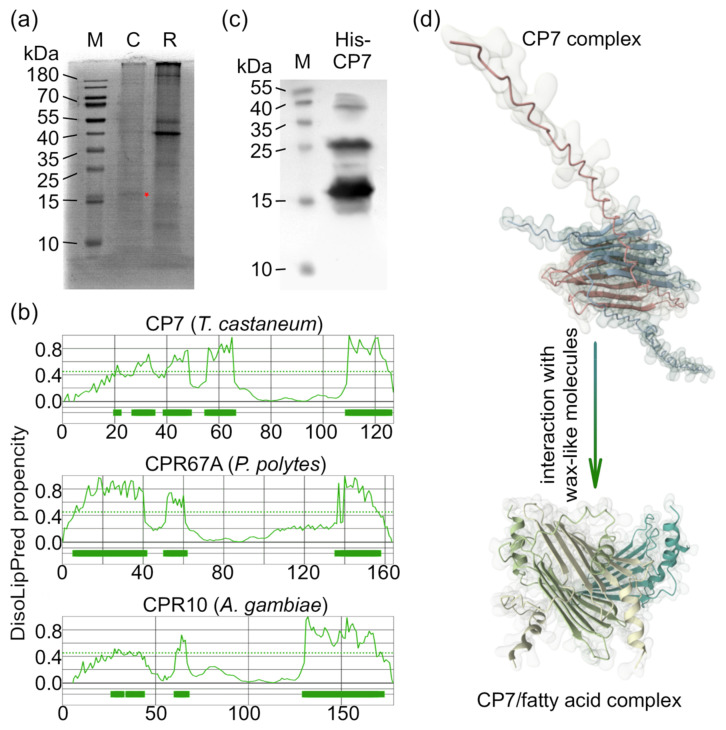
CP7 is a possible building block of Tenebrionoidea-like dimpled corneal nanocoatings. (**a**) SDS-PAGE analysis of corneal (C) and retinal (R) samples with CP7 protein band, used for the mass-spectrometry analysis, indicated by the red star. (**b**) DisoLipPred algorithm predicts extensive disordered lipid-binding regions in CP7, CPR67A, and CPR10. Thick lines show disordered liid-binding regions. (**c**) Western blot of purified RGS-His-CP7 shows the presence of stable oligomers in solution. (**d**) The potential complexation of CP7 subunits (shown by colors) in the solution and their assembly into a more compact and likely stable complex forced by interaction with fatty acids. Predicted by AlphaFold (https://alphafoldserver.com, accessed on 23 January 2025).

**Figure 4 biomimetics-10-00163-f004:**
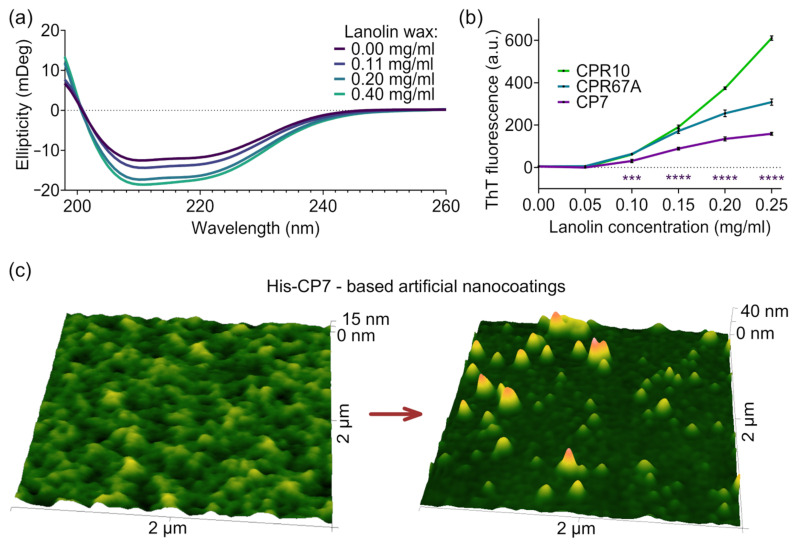
CP7 biophysical properties allow for the self-assembly of artificial nanocoatings. (**a**) Circular dichroism spectra of RGS-His-CP7 admixtures with different concentrations of the lanolin wax emulsion show protein folding. (**b**) ThT fluorescence measurements of proteins mixed with a lanolin emulsion at different concentrations. Data are shown as mean ± s.d, n = 4. Statistical significance compared to the control without wax emulsion was determined using a two-tailed *t*-test (“***” *p* < 0.001, “****” *p* < 0.0001). (**c**) Nanocoatings produced by using His-CP7 with the lanolin wax resemble corneal nanocoatings of *T. castaneum* (**left**) and can be modified by applying a self-assembling wax–protein mixture (**right**).

## Data Availability

All data generated or analyzed during this study are included in this published article and its Appendix A.

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
