# Peer review of "Smart Bio-Nanocoatings with Simple Post-Synthesis Reversible Adjustment"

_biomimetics, 2025, doi:10.3390/biomimetics10030163_

Round 1
Reviewer 1 Report
Comments and Suggestions for Authors
Please see attachment.

Author Response
The authors studied the structural and functional properties of corneal nanocoatings in beetles, focusing on how these coatings are modified and self-assembled to enhance anti-reflective and anti-contamination properties, and identified a novel protein, CP7, involved in the process. The article is an excellent study and the reported results could be useful to researchers in this area. I strongly recommend its publication after a suitable revision.
Thank you very much for the recommendation.
Method Section:
- It would be better to divide the Methods section into subsections, with each subsection explaining each technique in detail.
We agree with this comment. We have divided the section into better comprehensible subsections.
- There is no mention of how many Coleoptera samples were used, what is the sample size?
Sorry for the absence of this information. We added a minimal number of unique samples of each species used in this study.
- The method of "cleaning to remove contaminants" is not clear. What specific cleaning agents or method were used?
Thank you for pointing this out. There was a misleading sentence, which we changed. Washing was done only for the wettability measurement in the absence of "modifications".
- After cleaning and mounting, how were the samples stored before analysis?
We included this information. (Closed Petri dishes)
- What were the cantilever specifications for tapping/contact mode? What is the size of tip?
The subsection was expanded, and the required information was provided.
- The methodology states that before loading the sample on SDS-PAGE, it was boiled for 2 hours in the loading buffer. This duration is unusually long and could lead to protein degradation. Could you clarify the rationale behind this step or cite relevant literature that supports this protocol?
This protocol was established by us earlier. We changed the misleading word "boiled" to the better fitting phrase "incubated at 95°C". A shorter time of incubation leads to a low amount of protein extraction from the sample. Our experience also showed that even longer (up to 6 hours) incubation never led to the visible on SDS PAGE Coomassie protein degradation (this was done with pure recombinant proteins).
- For Western blot, blocking was performed using 5% low-fat milk in TBS, but this may not be ideal for all proteins. Why was TBST not used instead?
5% low-fat milk in TBS is a standard procedure for reducing the background signal on the membrane, which works with most protein-containing samples. In this case, the usage of the TBS was sufficient. Work with purified proteins does not require the addition of Tween (TBST), since there is no background signal. However, for RGS-His antibody, both protocols (TBS or TBST) would be suitable.
- In the Wax Emulsions section, the sonication time of 2 hours at 80°C seems excessive and could degrade some components. If this protocol is based on prior studies, please cite the relevant references.
Thank you for this comment. We not only included citations but also clarified the technique used. Indeed, sonication by tip for such a prolonged time is too harsh for the sample. In our case, we used an ultrasonic bath, which has a much milder effect.
- The method mentions an "upper layer enriched in wax nanodroplets." How was this enrichment confirmed?
This enrichment was confirmed visually due to clear phase separation. However, the final verification was done using a ZetaSizer Nanoseries (Malvern) analyser and DTS 1070 cells (Malvern) according to the manufacturer's protocol. This information was added to the manuscript.
Results Section:
- The study suggests that structural changes result from directed contamination due to higher adhesion in grooves. Was the chemical composition of the contaminants analyzed (e.g., via spectroscopy or chromatography) to confirm their nature?
Unfortunately, we did not conduct this investigation. The small amount of material makes this analysis very tricky. The only option that must work in this case is the Raman AFM, which is not available to us at the moment. However, these experiments are planned to be done in the future.
- The self-assembling wax-protein mixture was applied to T. castaneum cornea, leading to protrusion formation. What was the composition of this mixture (wax-to-protein ratio, solvent, incubation time)?
Clarification was introduced in methods (2.10 Nanocoatings generation)
- What mass spectrometry approach was used (e.g., LC-MS/MS, MALDI-TOF)? (In the caption of Fig.3.)
Subsection 2.4 Mass-spectrometry was expanded, and required information was added.
- The reported excitation (355 nm) and emission (535 nm) for ThT fluorescence are highly unusual. ThT excites around 440 nm and emits around 480 nm when bound to β-sheets. Could you explain why this excitation/emission was used, or cite a similar study that used this approach?
Indeed, we were forced to use these filters due to the significant light leakage in the instrument. Careful screening of all possible options gave us the best results with clear concentration dependency only when reported filters were used. We added the nominal bandpasses of the filters used in the methods.
Discussion:
- The authors suggest that these coatings provide anti-reflective properties but also imply potential anti-contamination properties, similar to those found in whiteflies. Have any controlled experiments been conducted to test whether these coatings repelcontaminants or dust?
Thank you for this question. No, such experiments were not done. We rewrote the sentence to make it more obvious.
- The study includes multiple beetle species, but it does not explain why different species have different coatings, despite their shared evolutionary history. Do these structural differences correlate with specific ecological factors, such as habitat, diet, or activity patterns (diurnal/nocturnal)?
We still do not know how to answer this essential question. There is definitely a correlation with some factors, but it seems that in each case, these factors are different. Some solutions appeared many times independently, while others are scarce. As we wrote in the Disscussion, these exact structures can be primordial, and it is possible that they appeared in the earlier stages of the evolution process, fulfilling the needs of different arthropods due to their bi-functionality.
Reviewer 2 Report
Comments and Suggestions for Authors
This study analyzed Coleoptera corneal nanocoatings and identified dimpled nanostructures, which can transform into maze-like/nipple-like protrusions, and identified cuticular protein 7 (CP7) was a key component of dimpled nanocoatings, demonstrating CP7's potential for bioinspired nanocoatings and provide insights into the evolutionary mechanisms of nanostructure formation. However, it has some questions that need to be improved.
1. In the Introduction part, there are little reviews of the relevant literature about the current nanopattern technologies, so it needs to add some comparative analysis of the existing technologies.
2. In the Results section, the resolution of some figures need to be improved, especially when presenting AFM images (Fig. 1a,c; Fig. 2d, Fig. 4c) and simulation results, so higher resolution images are recommended.
3. In the Results section, although AFM and other experimental data are provided, the interpretation and discussion of the data can be more in-depth, especially about the influence of different nanostructures on light reflection and surface wettability.
4. In the Discussion section, the reversible regulation mechanism of the nano-coating proposed in this paper is not clear enough. It is suggested to further elucidate the specific mechanism in the self-assembly process and how to achieve reversible regulation of the coating.
5. In the Conclusions section, please specify the main findings, the novelty and the limitations of this work.
Author Response
This study analyzed Coleoptera corneal nanocoatings and identified dimpled nanostructures, which can transform into maze-like/nipple-like protrusions, and identified cuticular protein 7 (CP7) was a key component of dimpled nanocoatings, demonstrating CP7's potential for bioinspired nanocoatings and provide insights into the evolutionary mechanisms of nanostructure formation. However, it has some questions that need to be improved.
- In the Introduction part, there are little reviews of the relevant literature about the current nanopattern technologies, so it needs to add some comparative analysis of the existing technologies.
We concur with this observation. Consequently, we have incorporated approximately ten citations from relevant studies and have included existing technologies (lines 42-46). However, we believe that a complete comparative analysis necessitates the format of a review article, given the significant increase in the amount of relevant literature in recent years.
- In the Results section, the resolution of some figures need to be improved, especially when presenting AFM images (Fig. 1a,c; Fig. 2d, Fig. 4c) and simulation results, so higher resolution images are recommended.
Thank you for highlighting this problem. Higher-resolution images are downloaded as separate files.
- In the Results section, although AFM and other experimental data are provided, the interpretation and discussion of the data can be more in-depth, especially about the influence of different nanostructures on light reflection and surface wettability.
We have included essential details that enrich the understanding of the results obtained. This added information provides clarity and insight, allowing for a deeper comprehension of the findings (lines 184-195; 210-218).
- In the Discussion section, the reversible regulation mechanism of the nano-coating proposed in this paper is not clear enough. It is suggested to further elucidate the specific mechanism in the self-assembly process and how to achieve reversible regulation of the coating.
We have illuminated the intricate mechanism underlying the process as a Reaction-Diffusion phenomenon. Furthermore, we have suggested that restricting the formation of nanostructures that adhere to the surface may offer a promising pathway to achieve effortless reversibility (lines 345-348).
- In the Conclusions section, please specify the main findings, the novelty and the limitations of this work.
Since the proposed mechanism remains uncharted territory, the entirety of the Conclusion section revolves around its groundbreaking nature. We included a dedicated paragraph that emphasizes our pivotal discovery, while also addressing the constraints of our study, such as limited wear resistance and a less-than-optimal adhesion to the substrate (lines 369-372; 378-379).
Round 2
Reviewer 2 Report
Comments and Suggestions for Authors
I think the manuscript has been sufficiently improved and can be accepted in current form.